# COVID-19 and cause of pregnancy loss during the pandemic: A systematic review

Seyyedeh Neda Kazemi[1]*, Bahareh Hajikhani[2], Hamidreza Didar[3], Sareh Sadat Hosseini[2], Sara Haddadi[4], Farima Khalili[3], Mehdi Mirsaeidi[4], Mohammad Javad Nasiri[2]*

**1** Preventive Gynecology Research Center, Shahid Beheshti University of Medical Sciences, Tehran, Iran, **2** Department of Microbiology, Shahid Beheshti University of Medical Sciences, Tehran, Iran, **3** School of Medicine, Shahid Beheshti University of Medical Sciences, Tehran, Iran, **4** Department of Medicine, Division of Pulmonary and Critical Care, University of Miami Miller School of Medicine, Miami, FL, United States of America

* s.nedakazemi@gmail.com (SNK); mj.nasiri@hotmail.com (MJN)

**Data Availability Statement:** All relevant data are within the manuscript and its Supporting Information files.

## Abstract

### Introduction

The association between Coronavirus Disease 2019 (COVID-19) and abortion has been debated since the beginning of the COVID-19 pandemic. We aimed to conduct this systematic review to understand better the potential effects of Severe Acute Respiratory Syndrome Coronavirus 2 (SARS-CoV-2) on fetal loss in infected mothers presented with abortion following this infection.

### Methods

We included articles published in PubMed/Medline, Web of Science, clinicaltrials.gov, and Embase databases in 2019 and 2020 through a comprehensive search via appropriate keywords, including COVID-19 and abortion synonyms. All studies with the abortion data in COVID-19 confirmed pregnant females were collected.

### Results

Out of 208 potentially relevant articles, 11 articles were eligible to include in the systematic review. The included reports were published because of the following reasons: (1) First-trimester miscarriage; (2) Late miscarriage; (3) complication of COVID-19 infection in pregnancy; (4) COVID-19 disease in artificial pregnancy. First-trimester abortion was found in 5 studies, and second-trimester abortion in 7 studies. Two patients acquired infection during the hospital stay while they were referred for abortion. Reports related to abortion in pregnant females with COVID-19 show that most miscarriages due to COVID-19 in the first trimester were due to placental insufficiency.

### Conclusions

There is an increased risk of abortion in mothers with a positive test result of SARS-CoV-2, which several case reports and case series have identified during the pandemic. Placental inflammation during the viral infection may result in fetal growth retardation and induce

**Funding:** The funders had no role in study design, data collection and analysis, decision to publish, or preparation of the manuscript.

**Competing interests:** The authors have declared that no competing interests exist.

abortion. There has not been any consistent evidence of vertical transmission of the virus from mother to fetus, which requires further investigation.

## Introduction

The first case of Coronavirus disease 2019 (COVID-19) was reported in Wuhan, China, in December 2019, and within a few months it developed into a worldwide pandemic [1]. COVID-19 is caused by severe acute respiratory syndrome coronavirus-2 (SARS-CoV-2). Pregnancy is considered a risk factor for severe illness from Coronavirus Disease 2019 (COVID-19) caused by Severe Acute Respiratory Syndrome Coronavirus 2 (SARS-CoV-2) infection [2–6]. Additionally, pregnant women with COVID-19 might be at increased risk for other adverse outcomes, such as preterm birth and abortion [6–10]. Studies have shown growing evidence related to COVID-19 association with abortion [11–13]. Transmission of this virus from infected mother to fetus has been debated since the beginning of the COVID-19 pandemic [14–17]. This controversy about the virus's vertical transmission has been studied in pregnant females experiencing COVID-19 signs and symptoms [14–17]. Vertical maternal-fetal transmission with severe fetal consequences may occur with maternal infection with TORCH agents (Toxoplasma, Other, Rubella, Cytomegalovirus, Herpes) and Zika virus [18–20]. As the fetal organs develop during the first trimester of pregnancy, maternal infections at this stage may be more severe compared to later gestational ages [18, 19]. Parvovirus B19 infection in first-trimester pregnancy, even in asymptomatic women, has been associated with an increased nuchal translucency thickness [21]. During the first 24 weeks, pregnancy loss is defined as abortion or miscarriage, which counts most pregnancy loss cases [22]. First-trimester miscarriage occurs in an estimated 10% to 15% of clinically diagnosed pregnancies. Risk factors and etiologies for first-trimester miscarriage are genetic, environmental, or multifactorial [23]. Mid-trimester pregnancy loss (MTL) occurs between 12 and 24 weeks gestation, and the estimated incidence rate has been noted to be 1% to 2% of pregnancies. The cause of MTI is often heterogeneous and And more than the first-trimester abortion is affected by the general conditions of the mother [24].

It is inferred that a significant factor related to abortion in mothers with COVID-19 is inflammation and placental insufficiency due to the direct effect of the virus on the placenta [3, 6]. Therefore, fetal death can be an outcome of COVID-19 in pregnancy [25]. There has been no systematic review about the studies reporting abortion in SARS-CoV-2 infected pregnant females. During the COVID-19 pandemic, abortions in mothers infected with SARS-CoV-2 are essential in determining effective preventive and therapeutic measures. The well-being of both mother and fetus requires further investigation of the effects of the SARS-CoV-2 virus during pregnancy.

## Methods

The present systematic review conforms to the "Preferred Reporting Items for Systematic Reviews and Meta-Analyses" (PRISMA) statement (S1 Table) [26].

### Search strategy

A comprehensive systematic search for relevant articles published in PubMed/Medline, Web of Science, clinicaltrials.gov, and Embase was conducted. All studies published in English up to October 30, 2020, were considered.

The following terms were used in the search strategy: Covid-19, severe acute respiratory syndrome coronavirus 2, SARS-CoV-2, in combinations with abortion synonyms. MeSH and Emtree terms were used when we searched in PubMed/Medline and Embase. Furthermore, references of selected papers were manually searched for additional relevant articles.

## Study selection

We evaluated all studies that reported data about abortion in pregnant patients with confirmed COVID-19. Abstracts, commentary, letters to editor, guidelines, and review articles were excluded.

Two investigators reviewed the titles and abstracts of the articles found during the initial search for relevance to the study inclusion criteria. All potentially relevant reports were obtained and thoroughly analyzed. Two investigators evaluated the full texts of articles independently, and any discrepancies were resolved through discussion.

## Data extraction and quality assessment

Data such as country of origin, the number of pregnant patients with confirmed COVID-19, clinical symptoms, laboratory findings, outcomes, diagnostic methods, and treatment were extracted from finally selected articles. Two authors extracted data from all included studies independently and created a data extraction form on an Excel sheet. Disagreements between authors were settled through discussions.

## Results

As illustrated in Fig 1, this search strategy resulted in an initial number of 208 potentially relevant articles, of which 132 were excluded by title and abstract evaluation. After applying the inclusion/exclusion criteria to the full-text documents, 11 articles, 5 case reports, and 6 case series from 9 countries were included with a total number of 196 unique cases of COVID-19 with a mean age of 30 years. All included studies were completely about pregnant patients in which abortion was mentioned as one of the complications of pregnancy (Table 1).

The included case reports/series were published because of the following reasons: (1) First-trimester miscarriage; (2) Late miscarriage; (3) complication of COVID-19 infection in pregnancy; and (4) COVID-19 infection in artificial pregnancy. Positive SARS-CoV-2 polymerase chain reaction (PCR) was present in all patients; however, computed tomography (CT) scan and chest x-ray (CXR) was also done in only two cases which indicated the COVID-19 pattern.

There were no comorbidities detected, such as hypertension, diabetes, cardiovascular disease, and pulmonary disease. Obesity was the only comorbidity in 4 studies, and the mean body mass index (BMI) was 29. One of the case reports was related to a patient with hemoglobin sickle cell disease. The induced abortion, in this case, was due to the fear of disease crises in the COVID-19 pandemic. This was the only case of abortion due to maternal safety when the mother acquired COVID-19 in the hospital. There was not any induced abortion due to fetal anomaly.

Patients had various symptoms, and the most common symptoms were cough and fever (Table 2). Lymphopenia and leukocytosis were reported in 2 studies, anemia and elevated C-Reactive Protein (CRP) in 3 studies (Table 3). Progression to acute respiratory distress syndrome (ARDS) was reported in 2 patients. One case with cardiomyopathy ended up being admitted to the intensive care unit (ICU) and had mortality due to COVID-19. A wide range of therapeutic modalities was tried across studies, and we summarize all of the drugs used (Table 4).

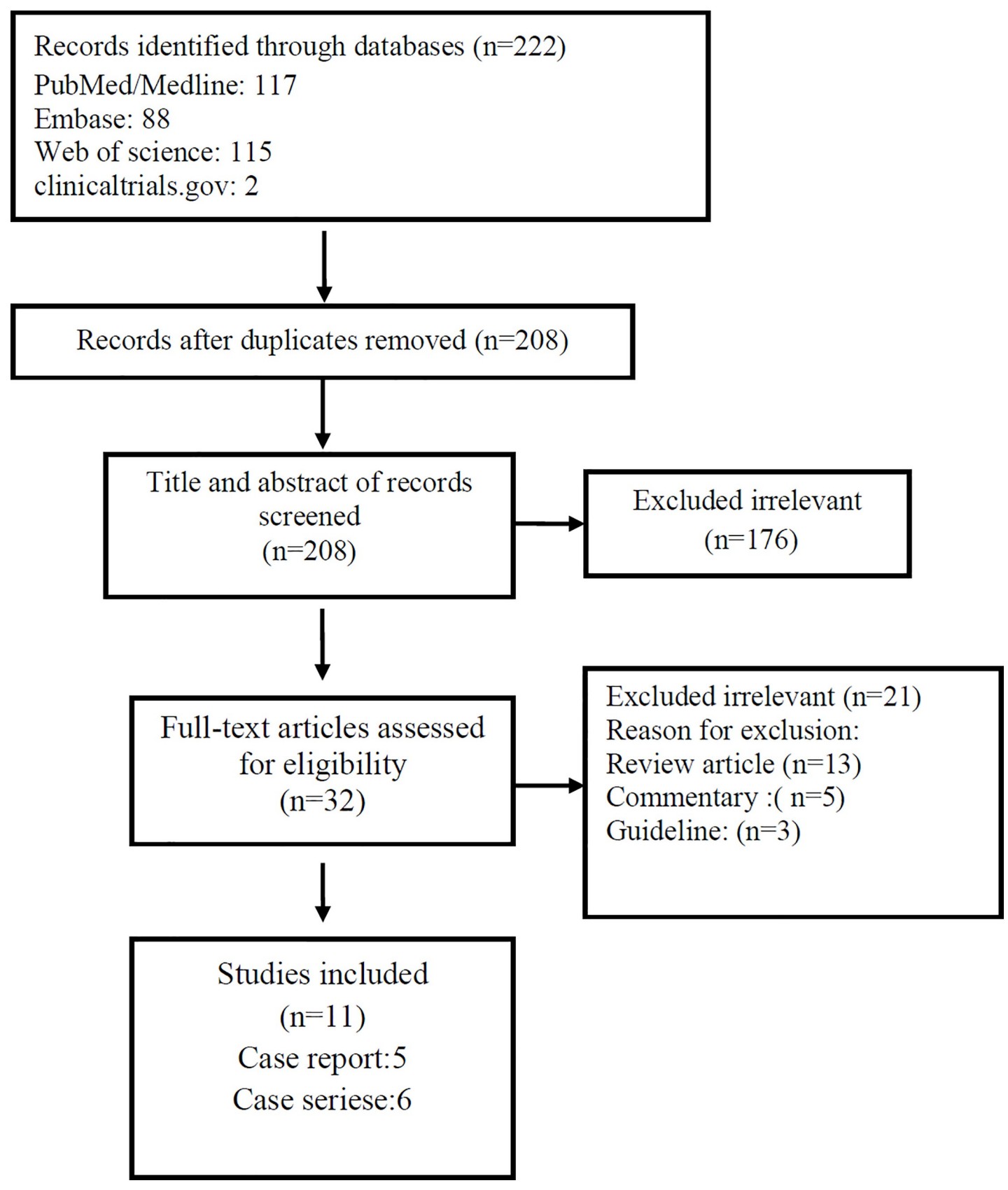

**Fig 1. Flow chart of study selection for inclusion in the systematic review.**

**Table 1. Characteristics of the included studies.**

| References | Country | Type of study | No. of mothers with COVID-19 | No. of abortion | Age | COVID-19 detection method |
|---|---|---|---|---|---|---|
| Fang, Nz. [41] | USA | Case report | 1 | 1 | 33 | PCR |
| Rana, M. S. [27] | Pakistan | Case report | 1 | 1 | 30 | PCR |
| Hachem, R. [42] | France | Case report | 1 | 1 | 21 | PCR |
| Baud, D. [29] | Switzerland | Case report | 1 | 1 | 28 | PCR |
| Shojaei, S. [43] | Iran | Case report | 1 | 1 | - | PCR |
| Wong, T. C. [39] | Malaysia | Case series | 2 | 2 | 34 | PCR |
| Yan, J. [44] | China | Case series | 116 | 1 | Mean age: 30 | PCR |
| Buonsenso, D. [45] | Italy | Case series | 7 | 1 | - | PCR |
| Richtmann, R. [25] | Brazil | Case series | 2 | 2 | Mean age: 29 | PCR |
| Mayeur, A. [28] | France | Case series | 10 | 1 | | PCR |
| Sentilhes, L. [46] | France | Case series | 54 | 1 | Mean age: 30 | PCR |

As indicated in Table 5, first-trimester abortion was found in 5 studies, and second-trimester abortion in 7 studies. Eight patients needed hospital admission for abortion, and one patient had an abortion at home. Two patients acquired infection during the hospital stay while they were referred for abortion. Four patients needed induction of abortion by prostaglandin, and two patients underwent curettage. Four studies showed fetal death before delivery in the second trimester. In 3 other studies, premature contraction caused the premature rupture of the membrane in one case and premature delivery in two other cases. Three studies showed placental inflammation based on the pathology report; however, only one study, SARS-CoV-2 PCR test, was positive [27–29]. Table 6 showed underlying conditions among patients with confirmed COVID-19 in the included studies.

**Table 2. Clinical manifestations in the included studies.**

| Variables | No of study | n/N* | % |
|---|---|---|---|
| Chest pain | 1 | 1/1 | 100 |
| Dyspnea | 4 | 6/128 | 4.68 |
| Shortness of breath | 4 | 33/172 | 19.18 |
| Cough | 9 | 81/188 | 43.08 |
| Fever | 7 | 83/185 | 44.86 |
| Sore throat | 4 | 35/172 | 20.34 |
| Fatigue | 4 | 72/172 | 41.86 |
| Myalgia | 4 | 13/128 | 10.15 |
| Abdominal pain | 1 | 1/1 | 100 |
| Nausea and vomiting | 1 | 1/1 | 100 |
| Diarrhea | 4 | 12/172 | 6.97 |
| Headache | 2 | 6/11 | 54.54 |
| Loss of taste | 3 | 24/65 | 36.92 |
| Loss of smell | 3 | 24/65 | 36.92 |
| Edema | 1 | 1/1 | 100 |
| Hypotension | 2 | 2/2 | 100 |
| Tachypnea | 1 | 1/1 | 100 |

*n: number of patients with any variable, N: Total number of mothers with COVID-19.

**Table 3. Laboratory and imaging findings in the included studies.**

| Variables | No of study | n/N* | % |
|---|---|---|---|
| Leukocytosis | 2 | 2/3 | 66.66 |
| Anemia | 3 | 3/118 | 2.54 |
| Lymphopenia | 4 | 77/172 | 44.76 |
| Involvement in CT | 4 | 118/172 | 68.6 |
| Bilateral involvement in CXR | 2 | 2/2 | 100 |

*n: number of patients with any variable, N: Total number of mothers with COVID-19.

## Discussion

Several observational studies on the possibility of vertical transmission of SARS-CoV-2 during pregnancy were published [30]. However, there is no evidence of transplacental transmission of COVID-19 to the fetus in the pregnancies [16]. Accordingly, it is essential to note that the fetus's immunoglobulin production may not occur during the first weeks of gestational age in the first trimester. Thus, this finding makes it difficult to understand whether SARS-CoV-2 could transmit vertically without any primary antibody response during the early gestation weeks [31]. In any case, further investigations are needed in first-trimester abortions and the babies born from mothers with IgM against SARS-CoV-2 [32].

The etiologic factors of the fetal loss during the first 24 weeks of gestation were classified into several systematic inflammatory events, including but not limited to the systematic inflammation involving the placenta [3]. These etiologic factors could cause premature contraction and premature rupture of the membrane resulting in premature delivery. In addition, it is well known that early pregnancy loss is predominantly related to intrinsic embryonic inborn errors [33]. In non-hospitalized Pregnant women with COVID-19 infection detected by serologic methods did not significantly differ in nuchal translucency thickness and double test in first-trimester screening. Furthermore, there was no significant increased risk of pregnancy loss in women with COVID-19 infection in the first trimester [34, 35]. Therefore, studies related to abortion in pregnant females with COVID-19 show that most of the miscarriages due to COVID-19 in the first trimester were due to placental insufficiency [36, 37]. Such conditions that led to the COVID-19 related miscarriages include spontaneous miscarriage, preterm delivery, and intrauterine growth restriction. The latter was also significantly higher during pregnancy in SARS-CoV. During the 2002–2003 epidemic, 57% of miscarriages in the first-trimester due to SARS-CoV have been documented [38]. Likewise, the Middle East

**Table 4. Treatment modalities in the included studies.**

| Variables | No of study | *n/N | % |
|---|---|---|---|
| Intubation | 3 | 6/171 | 3.5 |
| Hemoperfusion | 1 | 4/54 | 7.4 |
| Hydroxychloroquine | 4 | 4/5 | 80 |
| Steroid | 1 | 37/116 | 31.89 |
| Ritonavir | 3 | 5/56 | 8.92 |
| Lopinavir | 3 | 5/56 | 8.92 |
| IVIG | 1 | 1/1 | 100 |
| Azithromycin | 2 | 2/2 | 100 |

*n: number of patients with any variable, N: Total number of mothers with COVID-19.

**Table 5. The abortion time and procedure related to abortion management related to COVID-19.**

| Variables | No of study | n/N* | % |
|---|---|---|---|
| First trimester miscarriage | 4 | 5/126 | 3.96 |
| Second trimester miscarriage | 6 | 7/16 | 43.75 |
| Prostaglandin induced abortion | 3 | 4/4 | 100 |
| Spontaneous abortion | 8 | 10/140 | 7.14 |
| Curettage needed | 2 | 2/3 | 66.66 |
| Needed hospitalization | 7 | 8/9 | 88.88 |
| Abortion at home | 1 | 1/2 | 50 |
| COVID-19 after abortion | 2 | 2/2 | 100 |

*n: number of patients with any variable, N: Total number of mothers with COVID-19.

respiratory syndrome coronavirus (MERS-CoV) epidemic revealed higher rates of adverse pregnancy outcomes [27, 38, 39]. Acute or chronic placental insufficiency resulting in subsequent miscarriage or Intrauterine growth restriction (IUGR) was observed in 40% of maternal MERS-CoV and SARS-CoV [29]. Recent studies have shown the deposition of perivillous fibrin and multiple villous infarcts in the placenta of SARS-CoV-2 infected mothers [27]. As a result, the placental infection can disturb the transportation of nutrients from mother to fetus, leading to adverse pregnancy outcomes [27].

Some newborns have tested positive for SARS-CoV-2 shortly after birth [40]. It is unknown if these newborns were infected by the virus before, during, or after birth from close contact with an infected person. Different routes of mother to fetus SARS-CoV-2 transmission, including vertical transmission and the placenta dysfunction/abortion, need to be carefully investigated in clinical and animal studies.

This study has some limitations, including the limited data during the pandemic in pregnant females. This could be partially related to pregnant mothers' fear of being visited in clinics during routine prenatal care in the COVID-19 pandemic. Moreover, lack of pathology results and SARS-CoV-2 PCR test and limited sampling from the aborted fetus or placenta due to the lack of consent from the mother, etc., has kept the current data of these abortions limited to the cases reported.

In conclusion, there is an increased risk of abortion in mothers who tested positive for SARS-CoV-2, which case reports and case series have identified during the pandemic. Most of these studies referred to the effect of the virus on the placenta and inflammation, resulting in fetal growth retardation and may induce abortion. Data from the pregnant females during the pandemic who experienced abortion is limited. Therefore, further investigation is needed to identify the reasons behind pregnancy loss in SARS-CoV-2 infected mothers and whether

**Table 6. Underlying condition in the included studies.**

| Variables | No of study | n/N* | % |
|---|---|---|---|
| History of obesity | 4 | 8/56 | 5.37 |
| History of hypertension | 2 | 6/170 | 3.52 |
| History of gestational diabetes | 2 | 13/170 | 7.64 |
| History of preeclampsia | 3 | 8/177 | 4.51 |

*n: number of patients with any variable, N: Total number of mothers with COVID-19.

there could be any evidence of the viral transmission to the fetus causing abortion due to the virus's direct effect on the fetal organs.

## Supporting information

**S1 Table. PRISMA checklist.**
(DOCX)

## Author Contributions

**Conceptualization:** Seyyedeh Neda Kazemi, Mehdi Mirsaeidi, Mohammad Javad Nasiri.

**Data curation:** Seyyedeh Neda Kazemi, Bahareh Hajikhani, Hamidreza Didar, Sareh Sadat Hosseini, Sara Haddadi, Farima Khalili, Mohammad Javad Nasiri.

**Formal analysis:** Bahareh Hajikhani, Mohammad Javad Nasiri.

**Investigation:** Mohammad Javad Nasiri.

**Methodology:** Mohammad Javad Nasiri.

**Project administration:** Mohammad Javad Nasiri.

**Resources:** Mohammad Javad Nasiri.

**Software:** Mohammad Javad Nasiri.

**Supervision:** Mehdi Mirsaeidi, Mohammad Javad Nasiri.

**Validation:** Mohammad Javad Nasiri.

**Writing – original draft:** Bahareh Hajikhani, Mohammad Javad Nasiri.

**Writing – review & editing:** Mehdi Mirsaeidi, Mohammad Javad Nasiri.

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
