## [Decision Letter · Decision Letter 0]

19 Jul 2021

PONE-D-21-13530

COVID-19 and cause of pregnancy loss during the pandemic: A systematic review

PLOS ONE

Dear Dr. Nasiri,

Thank you for submitting your manuscript to PLOS ONE. After careful consideration, we feel that it has merit but does not fully meet PLOS ONE’s publication criteria as it currently stands. Therefore, we invite you to submit a revised version of the manuscript that addresses the points raised during the review process.

ACADEMIC EDITOR:

Please revise according to reviewer's suggestion before forwarding to publication.

Thank you for your patience for long time.

We look forward to receiving your revised manuscript.

Kind regards,

Kazumichi Fujioka

Academic Editor

PLOS ONE

Journal Requirements:

"The funders had no role in study design, data collection and analysis, decision to

publish, or preparation of the manuscript."

Reviewers' comments:

Reviewer's Responses to Questions

**Comments to the Author**

1. Is the manuscript technically sound, and do the data support the conclusions?

Reviewer #1: Yes

Reviewer #2: Yes

2. Has the statistical analysis been performed appropriately and rigorously? 

Reviewer #1: Yes

Reviewer #2: Yes

3. Have the authors made all data underlying the findings in their manuscript fully available?

Reviewer #1: Yes

Reviewer #2: Yes

4. Is the manuscript presented in an intelligible fashion and written in standard English?

Reviewer #1: Yes

Reviewer #2: Yes

5. Review Comments to the Author

Reviewer #1: Dear Respectable Editor

Thank you to give me the opportunity to review the article entitled (COVID-19 and cause of pregnancy loss during the pandemic: A systematic review)

The study designed to understand better the potential effects of Severe Acute Respiratory Syndrome Coronavirus 2 (SARS-CoV-2) on fetal loss in infected mothers presented with abortion following this infection.

The study concluded that there is an increased risk of abortion in mothers with a positive test result of SARS-CoV-2. Placental inflammation during the viral infection may result in fetal growth retardation and induce abortion. There has not been any consistent evidence of vertical transmission of the virus from mother to fetus, which requires further investigation.

Authors provided structured abstract and informative title

In the introduction, thy mentioned the controversy between studies regarding COVID-19 and explained why their study was designed (there is no systematic review reporting abortion in SARS-CoV-2 infected pregnant females).

In the methods section, they mentioned the search strategy, study selection, and data extraction and quality assessment.

Authors provided clear results, and a conclusion reflecting their results.

In addition, authors wrote the limitations of their study, future suggested research, and they used relevant references.

It my opinion its interesting study and can be accepted for publication without suggested changes from my side.

Regards

Reviewer #2: Dear authors,

I appreciate your efforts to work on COVID-19. The manuscript is well written. Suggestions to improve:

1) Please add more data to “Introduction” section.

2) Before using the abbreviation in main body of manuscript, please write it in full

3) Please choose the keywords from the MeSH database.

4) Please add more data to “Discussion” to describe your valuable work.

Best regards

6. PLOS authors have the option to publish the peer review history of their article (what does this mean?). If published, this will include your full peer review and any attached files.

Reviewer #1: **Yes: **Ibrahim A. Abdelazim (Professor of OB/GYN Ain Shams University, Egypt)

Reviewer #2: **Yes: **Mohsen Heiday

---

## [Author Response · Author response to Decision Letter 0]

26 Jul 2021

Dear Editor and reviewers,

Thank you for your considering our manuscript " COVID-19 and cause of pregnancy loss during the pandemic: A systematic review" We thank the editor and reviewers for their thoughtful critique and comments. We have carefully edited the manuscript as requested by reviewers and have provided a point-by-point response below. Please find the revised version included. We hope this meets the established reputation for the quality of your esteemed journal.

Thank you for considering our manuscript.

Corresponding author:

Mohammad Javad Nasiri, PhD, MPH 

Email: mj.nasiri@hotmail.com

Reviewer #1: Dear Respectable Editor

Thank you to give me the opportunity to review the article entitled (COVID-19 and cause of pregnancy loss during the pandemic: A systematic review)

The study designed to understand better the potential effects of Severe Acute Respiratory Syndrome Coronavirus 2 (SARS-CoV-2) on fetal loss in infected mothers presented with abortion following this infection.

The study concluded that there is an increased risk of abortion in mothers with a positive test result of SARS-CoV-2. Placental inflammation during the viral infection may result in fetal growth retardation and induce abortion. There has not been any consistent evidence of vertical transmission of the virus from mother to fetus, which requires further investigation.

Authors provided structured abstract and informative title

In the introduction, thy mentioned the controversy between studies regarding COVID-19 and explained why their study was designed (there is no systematic review reporting abortion in SARS-CoV-2 infected pregnant females).

In the methods section, they mentioned the search strategy, study selection, and data extraction and quality assessment.

Authors provided clear results, and a conclusion reflecting their results.

In addition, authors wrote the limitations of their study, future suggested research, and they used relevant references.

It my opinion its interesting study and can be accepted for publication without suggested changes from my side.

Regards

Response: Dear reviewer, thank you very much for your encouraging comments. 

Reviewer #2: Dear authors,

I appreciate your efforts to work on COVID-19. The manuscript is well written. Suggestions to improve:

1) Please add more data to “Introduction” section.

Response: Done as requested.

2) Before using the abbreviation in main body of manuscript, please write it in full

Response: Done as requested.

3) Please choose the keywords from the MeSH database.

Response: Dear reviewer, when we searched PubMed and Embase, we used MeSH and Emtree. The manuscript was accordingly edited. (methods section, lines: 79-80).

4) Please add more data to “Discussion” to describe your valuable work.

Response: Done as requested.

---

## [Editor Report · Decision Letter 1]

28 Jul 2021

COVID-19 and cause of pregnancy loss during the pandemic: A systematic review

PONE-D-21-13530R1

Dear Dr. Nasiri,

We’re pleased to inform you that your manuscript has been judged scientifically suitable for publication and will be formally accepted for publication once it meets all outstanding technical requirements.

Kind regards,

Kazumichi Fujioka

Academic Editor

PLOS ONE

Additional Editor Comments (optional):

I am grateful for your patience during COVID era when we experience difficulty finding enough competent reviewers timely.
---

## [Editor Report · Acceptance letter]

3 Aug 2021

PONE-D-21-13530R1 

COVID-19 and cause of pregnancy loss during the pandemic: A systematic review 

Dear Dr. Nasiri:

I'm pleased to inform you that your manuscript has been deemed suitable for publication in PLOS ONE. Congratulations! Your manuscript is now with our production department. 

Kind regards, 

on behalf of

Dr. Kazumichi Fujioka 

Academic Editor

PLOS ONE